# Sudden cardiac death after coronary artery bypass graft surgery and role of antiplatelet therapy

**Iftikhar Ali Ch**[1,2]*, **Azhar Chaudhry**[1], **Mishal Zehra**[1], **Pei-Tzu Wu**[1,3], **Mashal Tahirkheli**[1], **Poonam Bai**[1], **Ananya Bhaktaram**[1], **Rahat Jamal**[1], **Sabrina Chaudry**[1], **Faris Amil**[1], **John Randolph**[4], **Steven Miller**[4], **Jeffrey Garrett**[4], **Naeem Tahirkheli**[1,4]

**1** South Oklahoma Heart Research, Oklahoma City, Oklahoma, United States of America, **2** SSM Health Saint Anthony Hospital, Oklahoma City, Oklahoma, United States of America, **3** Doctor of Physical Therapy Program, Southern California University of Health Sciences, Whittier, California, United States of America, **4** Oklahoma Heart Hospital, Oklahoma City, Oklahoma, United States of America

* babar_175@hotmail.com

**Data availability statement:** Data availability statement The data supporting the findings of this study are available upon reasonable request. However, restrictions apply as these data were used under license for this study and cannot be made publicly accessible. The hospital's policy strictly prohibits sharing patient data online to avoid potential breaches of confidentiality and violations of applicable data protection laws. In accordance with these

## Abstract

### Background

While coronary artery bypass grafting (CABG) surgery is effective in reducing the risk of myocardial infarction and subsequent cardiac events by improving myocardial perfusion, the risk of sudden cardiac death (SCD) remains notable.

### Methods

This retrospective observational study evaluated the efficacy of dual antiplatelet therapy (DAPT) in preventing sudden cardiac death (SCD) among patients undergoing CABG surgery at a major U.S. cardiac center (2012–2015). Data was manually extracted from electronic medical records between 23/04/2017 to 30/03/ 2018 and verified for accuracy, with patients categorized into DAPT or aspirin monotherapy groups based on discharge prescriptions.

### Results

Of 2,476 patients followed in this post-CABG study, the analysis included 1,005 patients who received aspirin monotherapy (AMT) and 1,458 patients who received dual antiplatelet therapy (DAPT). AMT group had a significantly higher incidence of SCD compared to those on DAPT (3.1% vs 0.8%; OR=3.831, 95% CI: 1.961–7.519; p<0.001). The binary regression model indicated that a higher BMI was associated with an increased risk of SCD (HR=1.064, 95% CI: 1.012–1.118, p=0.014). However, patients prescribed P2Y12 antagonists (HR=0.285, 95% CI: 0.135–0.603, p<0.001), those with a GFR > 60 ml/min (HR=0.314, 95% CI: 0.158–0.624, p<0.001), and those with a higher ejection fraction (HR=0.962, 95% CI: 0.939–0.986, p=0.002) were less likely to experience SCD following CABG. A 1 kg/m2 increase in BMI is associated with a 6.4% increase in the risk of SCD. Morbidly obese patients with BMI > 35 were more likely to have experienced SCD than

regulations, the full dataset cannot be publicly shared. Nevertheless, researchers may request access to the data by contacting the authors and obtaining permission from Oklahoma Heart Hospital. We respectfully request consideration for exempt status based on these constraints. Patient data has been securely stored on the hospital's systems and is accessible to the research department and authorized personnel approved by the Medical Executive Committee. For any external requests related to data access, Mr. Navid Chaudry, Director of South Oklahoma Heart Research, can be contacted. His details are as follows: Navid Chaudry, MBA, CRC Email: nchaudry@okheart.com Fax: 405-628-6795 Address: 5224 E I-240 Service Road, Suite 201, Oklahoma City, OK 73135 Additionally, we ensure the persistent and long-term storage of data through hospital systems, which comply with institutional and regulatory standards for data security and availability.

**Funding:** The author(s) received no specific funding for this work.

**Competing interests:** The authors have declared that no competing interests exist.

**Abbreviations:** SCD, Sudden cardiac death; CABG, Coronary artery bypass graft; DAPT, Dual antiplatelet therapy; AMT, Aspirin mono-therapy; ICD, Implantable cardiac defibril-lator; CVA, Cerebrovascular accident; CV, Cardiovascular; OR, Odds ratio; CI, Confidence interval; PCI, Percutaneous coronary inter-vention; IRB, Institutional review board; EMR, Electronic medical record; RCT, Randomized controlled trial.

those with BMI < 35 (HR = 2.400, 95% CI: 1.204–4,787; p = 0.013). Similarly, patients with EF > 40% had decreased incidence of SCD compared to those with EF < 40% (HR 0.347, 95% CI:0.158–0.763; p = 0.008). Patients on AMT had higher all-cause (OR = 2.136, 95% CI 1.502–3.038; p < 0.001) and CV mortality (OR = 3.731, 95% CI 2.233–6.235; p < 0.001) but had lower incidence of major bleeding (by drop in hemoglobin criteria) (OR = 0.704, 95% CI: 0.595–0.833; p < 0.001) compared to the DAPT group.

## Conclusion

DAPT prescription after CABG improves survival and lowers risk of sudden cardiac death.

## Introduction

### Background

Sudden cardiac death (SCD) is characterized by an unexpected, natural death due to cardiac causes, typically occurring within one hour of the onset of symptoms [1]. SCD represents a significant public health concern due to its sudden onset and often unpredictable occurrence, impacting individuals across a wide range of ages and health conditions. The incidence of SCD varies significantly based on factors such as patient demographics, comorbidities, quality of surgical and postoperative care, and duration of follow-up.

### Current knowledge

While coronary artery bypass grafting (CABG) surgery is effective in reducing the risk of myo-cardial infarction and subsequent cardiac events by improving myocardial perfusion, the risk of SCD remains notable, with reported rates ranging from less than 1% to over 5% per year [2–5]. This persistent risk is likely due to factors such as incomplete or failed revascularization, pro-gression of native vessel disease, and the presence of myocardial scars, which can create a sub-strate for arrhythmogenic activity. A study analyzing a large database of patients who underwent percutaneous coronary intervention (PCI) or CABG identified several independent predictors of SCD in patients with chronic heart disease. These predictors included chronic obstructive pulmonary disease, congestive heart failure, reduced left ventricular ejection fraction (LVEF) ≤ 30%, chronic total occlusion of a coronary artery, diabetes requiring insulin therapy, chronic renal disease, and peripheral artery disease [5]. Thus, managing cardiovascular (CV) risk factors has remained the focus of preventive strategy for improving clinical outcomes after CABG.

The CABG Patch Trial further underscored the complexity of SCD prevention by demon-strating that while implantable cardioverter-defibrillator (ICD) therapy reduced arrhythmic deaths by 45%, it did not significantly affect non-arrhythmic deaths, resulting in no significant reduction in total mortality, as 71% of the deaths were non-arrhythmic [6]. A nationwide database from Sweden reported that 58% of all deaths following CABG were cardiac in origin, with heart failure being the most common cause (65% within 30 days post-CABG and 36% beyond 30 days), followed by myocardial infarction (56% and 29%, respectively) [7]. These two studies suggest failure of revascularization may be a bigger factor in this patient popula-tion leading to adverse clinical outcomes than arrhythmogenic substrate from scarring. Evi-dence has shown that CABG can reduce the incidence of SCD more effectively than PCI, with improved revascularization status and by preventing myocardial ischemia [8]. While patients with persistent low ejection fraction despite optimal GDMT are recommended to have ICD implant [9]; it is not recommended for patients undergoing CABG as a prophylaxis.

## Gap in knowledge

Complete surgical revascularization through CABG and optimal management of CV risk factors may be the most effective strategy for preventing fatal cardiac events leading to SCD. Therefore, it is vital to optimize medical therapy including antiplatelet therapy to sustain revascularization status after CABG. In recent years more evidence has emerged supporting dual antiplatelet therapy's (DAPT) clinical benefits in CABG patients [10–12]. DAPT's role in preventing acute graft occlusion, improving graft patency, and thereby maintaining revascularization and reducing the risk of fatal cardiac events could be of vital importance. Although most studies have not specifically evaluated the impact of DAPT on SCD and compared it with aspirin monotherapy, Kulik et al. demonstrated a reduced incidence of SCD in patients treated with ticagrelor compared to those receiving clopidogrel [12]. Recent literature suggests that DAPT could be an important and underexplored strategy in preventive medicine especially after surgical revascularization as compared to PCI.

## Hypothesis

In this study, we hypothesize that DAPT could potentially decrease the incidence of ischemic events and fatal myocardial infarction by enhancing graft survival thus lowering the risk of SCD after CABG. We sought to investigate the role of different antiplatelet therapies and clinical factors in preventing risk of SCD after CABG.

## Methods

### Study design

This retrospective observational study assessed the efficacy of DAPT in preventing SCD in patients undergoing elective or emergent CABG surgery at one of the largest cardiac surgery centers in the United States.

**Inclusion criteria.**

1. All CABG surgeries performed at Oklahoma Heart Hospital, Oklahoma City, OK between 2012–2015.

2. On and off pump CABG surgeries.

**Exclusion criteria.**

1. Patients who underwent concomitant valve surgery.

2. Need for long-term anticoagulation.

3. Aspirin intolerance,

4. Patients who did not follow up at the institute after surgery.

### Ethics statement

This ethics statement affirms that the research adhered to the highest ethical standards and guidelines established by regulatory bodies. Informed consent was waived by the Institutional Review Board. Confidentiality and privacy of participant information were maintained, conflicts of interest were denied, high morals were upheld, data integrity and transparency were ensured, authorship criteria were outlined, and contributions were acknowledged appropriately. Approval of this study was obtained from both the hospital's medical executive committee and the institutional review board (IRB). IRB approval was obtained from Western Institutional Review Board

Puyallup, WA 98374-2115, Study Num: 1174207, WO Num: 1-1004749-1. The board found that this research meets the requirements for a waiver of consent under 45 CFR 46.116(d).

## Data collection and access dates

A team of trained researchers manually reviewed electronic medical records (EMRs) between **23/04/2017 to 30/03/2018**, including clinical notes, operative reports, laboratory studies, and discharge medications. Data entry was double-checked by a second investigator to ensure accuracy. Additionally, medical records from other healthcare facilities were obtained and examined for relevant outcomes. Phone follow-up and interviews were conducted if necessary to gather missing information.

## Outcome measures

SCD was defined as sudden death due to cardiac causes within 1–2 hour of onset of symptoms. Cardiovascular (CV) mortality included deaths from various cardiac and vascular causes. Post-CABG acute coronary syndrome (ACS) was identified using appropriate ICD codes and was confirmed by EKG findings and abnormal laboratory values. Cerebrovascular accident (CVA) was defined as focal neurological deficits lasting at least 24 hours or leading to death. Major bleeding was defined as a drop in hemoglobin greater than 5 g/dl, hemorrhagic cardiac tamponade, intracranial hemorrhage, or any bleeding causing hemodynamic instability.

## Group assignment

Based on the antiplatelet therapy received at discharge, patients were categorized into two groups:

1. DAPT Group: Patients were classified as DAPT group if they were given prescription of combination of aspirin and a P2Y12 receptor antagonist after CABG.

2. Aspirin Monotherapy (AMT) Group: Patients were prescribed aspirin after CABG.

## Statistical analysis

Descriptive statistics and frequencies were calculated for continuous and categorical variables. Continuous variables were compared between groups using the Mann–Whitney U test. Categorical variables were compared between groups using the chi-squared test. Survival rates were compared between groups using Kaplan–Meier analysis. The Cox proportional-hazards regression model was estimated using the forward conditional method based on the likelihood ratio. The incidence proportions and odds ratios of various outcomes were compared between the AMT and DAPT groups. Statistical significance was set at a two-tailed p value, less than 0.05. All analyses were conducted with SPSS version 29.0 (IBM Corp., Armonk, NY).

## Results

### Patient demographics

Of 2,476 patients followed in this post-CABG study, the analysis included 1,005 patients who received AMT and 1,458 patients who received DAPT. AMT group had a higher prevalence of CKD (18.8% vs. 14.1%; p = 0.002) and stable angina (65.6% vs. 60.6%; p = 0.013) compared to those on DAPT. Conversely, patients on DAPT exhibited a higher prevalence of peripheral arterial disease (14.8% vs. 10.5%; p = 0.002), glomerular filtration rate (GFR) < 60 (44.98% vs.

42.59%; p = 0.020), prior PCI (24.5% vs. 20.1%; p = 0.011), pre-CABG use of DAPT (30.0% vs. 15.5%; p < 0.001), and were more likely to undergo on-pump surgery (79.2% vs. 68.5%; p < 0.001) with a longer duration (210.15 minutes vs. 185.66 minutes; p < 0.001). Post-CABG, the medication profile for DAPT patients indicated a higher prescription rate of statins (90.7% vs. 86.6%; p < 0.001) and antiarrhythmic drugs (AAD) (33.3% vs. 22.6%; p < 0.001) (for details, see Table 1).

**Table 1. Patient demographics and baseline characteristics.**

| | AMT | DAPT | p-value |
|---|---|---|---|
| | N = 1,005 | N = 1,458 | |
| Age (Years) | 65.67 ± 10.02 | 64.71 ± 10.02 | 0.019 |
| Sex (n of Male) | 761 (75.8%) | 1,060 (72.7%) | 0.086 |
| Race/Ethnicity (n) | | | 0.282 |
| Caucasian | 885 (88.1%) | 1,255 (86.1%) | |
| African American | 32 (3.2%) | 58 (4.0%) | |
| Native American | 37 (3.7%) | 67 (4.6%) | |
| Hispanic | 25 (2.8%) | 44 (3.0%) | |
| Asian | 13 (1.3%) | 18 (1.2%) | |
| Pacific Islander | 1 (0.1%) | 1 (0.1%) | |
| Multiple | 2 (0.2%) | 9 (0.6%) | |
| Unknown | 10 (1.0%) | 6 (0.4%) | |
| BMI (Kg/m$^2$) | 30.63 ± 5.78 | 30.75 ± 5.98 | |
| Smoker (n) | 516 (51.4%) | 816 (56.1%) | 0.620 |
| Co-morbidities (n) | | | |
| Hypertension | 873 (86.9%) | 1,268 (87.0%) | 0.941 |
| CVA | 70 (7.0%) | 131 (9.0%) | 0.072 |
| COPD | 253 (25.2%) | 393 (27.0%) | 0.323 |
| Hyperlipidemia | 707 (70.3%) | 1,029 (70.6%) | 0.903 |
| Diabetes Mellitus | 423 (42.1%) | 583 (40.0%) | 0.297 |
| CKD | 189 (18.8%) | 205 (14.1%) | 0.002 |
| MI | 287 (28.6%) | 461 (31.6%) | 0.104 |
| Stable Angina | 659 (65.6%) | 884 (60.6%) | 0.013 |
| CHF | 215 (21.4%) | 302 (20.7%) | 0.684 |
| PAD | 106 (10.5%) | 216 (14.8%) | 0.002 |
| HbA1c (%) | 6.63 ± 1.52 | 6.61 ± 1.51 | 0.773 |
| GFR | | | |
| > 60 ml/min/1.73 m$^2$ (n) | 705 (70.2%) | 1,052 (72.6%) | 0.208 |
| < 60 ml/min/1.73 m$^2$ (Avg) | 42.59 ± 13.80 | 44.98 ± 13.11 | 0.020 |
| Ejection Fraction (%) | 52.29 ± 11.55 | 54.44 ± 11.98 | < 0.001 |
| Previous PCI/Stent (n) | 202 (20.1%) | 357 (24.5%) | 0.011 |
| PCI | 188 (18.7%) | 329 (22.6%) | 0.021 |
| Stent | 107 (10.6%) | 194 (13.3%) | 0.048 |
| Pre-CABG Medications (n) | | | |
| ASA | 682 (67.9%) | 1,031 (70.8%) | 0.118 |
| P2Y12 receptor antagonist | 156 (15.5%) | 437 (30.0%) | < 0.001 |
| Surgical Variables | | | |
| On-Pump surgery (n) | 679 (68.5%) | 1,124 (79.2%) | < 0.001 |
| Surgery duration (min) | 185.66 ± 55.87 | 210.15 ± 70.27 | < 0.001 |

*(Continued)*

**Table 1.** (Continued)

| | AMT | DAPT | p-value |
|---|---|---|---|
| | N = 1,005 | N = 1,458 | |
| Post -CABG Medications (n) | | | |
| BB | 887 (88.3%) | 1,273 (87.3%) | 0.482 |
| ACE-I/ARB | 531 (52.8%) | 837 (57.4%) | 0.025 |
| AAD | 227 (22.6%) | 485 (33.3%) | < 0.001 |
| CCB | 165 (16.4%) | 277 (19.0%) | 0.101 |
| Statin | 870 (86.6%) | 1,322 (90.7%) | 0.001 |
| P2Y12 receptor antagonist | 0 | 1,458 (100%) | < 0.001 |
| Survival Duration (days) | 1,067.01 ± 552.18 | 1,040.42 ± 530.19 | 0.229 |

Values are n (%) or n, unless otherwise indicated.

AMT = aspirin monotherapy; DAPT = dual antiplatelet therapy; BMI = body mass index; CVA = cerebrovascular accident; COPD = chronic obstructive pulmonary disease; CKD = chronic kidney disease; MI = myocardial infarction; CHF = congestive heart failure; PAD = peripheral arterial disease; GFR = glomerular filtration rate; PCI = percutaneous coronary intervention; CABG = coronary artery bypass graft; ASA = aspirin; BB = beta-blocker; ACE-I = angiotensin-converting enzyme inhibitor; ARB = angiotensin receptor blocker; AAD = antiarrhythmic drugs; CCB = calcium channel blocker

## Sudden cardiac death and overall survival

After adjusting for the above variables, patients who received AMT after CABG had a significantly higher incidence of SCD compared to those on DAPT (3.1% vs 0.8%; OR = 3.831, 95% CI: 1.961–7.519; p < 0.001) (see Table 2). Patients on AMT had higher all-cause (OR = 2.136, 95% CI 1.502–3.038; p < 0.001) and CV mortality (OR = 3.731, 95% CI 2.233–6.235; p < 0.001) (Fig 1). AMT had lower incidence of major bleeding (OR = 0.704, 95% CI: 0.595–0.833; p < 0.001) compared to the DAPT group. However, the incidence of transfusion requirements was 9.9% (100 out of 1005) in the AMT group and 15.6% (159 out of 1458) in the DAPT group, with no significant difference between the two groups (OR: 1.11, 95% CI: 0.85–1.45, p = 0.48). There were no cases of cardiac tamponade reported in either the AMT or DAPT groups. However, one patient in the DAPT group died due to bleeding complications. Additionally, no cases of CNS bleeding were observed.

The cumulative survival rate is higher in the DAPT group as compared to the AMT group.

## Predictors of sudden cardiac death

The binary regression model indicated that a higher BMI was associated with an increased risk of SCD (HR = 1.064, 95% CI: 1.012–1.118, p = 0.014). However, patients prescribed P2Y12 antagonists (HR = 0.285, 95% CI: 0.135–0.603, p < 0.001), those with a GFR > 60 ml/min (HR = 0.314, 95% CI: 0.158–0.624, p < 0.001), and those with a higher ejection fraction (HR = 0.962, 95% CI: 0.939–0.986, p = 0.002) were less likely to experience SCD following CABG (see Table 3). A 1 kg/m2 increase in BMI is associated with a 6.4% increase in the risk of SCD, and a 1% increase in EF is associated with a 3.8% decrease in the risk of SCD (Fig 2). Morbidly obese patients with BMI > 35 were more likely to have experienced SCD than those with BMI < 35 (HR = 2.400, 95% CI: 1.204–4.787; p = 0.013). Similarly, patients with EF > 40% had a decreased incidence of SCD compared to those with EF < 40% (HR 0.347, 95% CI:0.158–0.763; p = 0.008) (Table 4). Common post-CABG medications, including angiotensin-converting enzyme inhibitors/angiotensin receptor blockers (ACE-I/ARBs), beta-blockers, antiarrhythmic drugs (AAD), calcium channel blockers (CCB), and statins, were not significant predictors of SCD (p = 0.380, 0.450, 0.428, 0.337, and 0.185, respectively).

**Table 2. Clinical outcomes.**

| N (%) | AMT | DAPT | OR | 95% CI | p-value |
|---|---|---|---|---|---|
| All-Cause Mortality | 79 (7.9%) | 56 (3.8%) | 2.136 | 1.502–3.038 | < 0.001 |
| SCD | 31 (3.1%) | 12 (0.8%) | 3.831 | 1.961–7.519 | < 0.001 |
| CV Mortality | 52 (5.2%) | 21 (1.5%) | 3.731 | 2.233–6.235 | < 0.001 |
| Post-CABG ACS | 39 (3.9%) | 79 (5.4%) | 0.706 | 0.477–1.046 | 0.081 |
| Post-CABG CVA | 27 (2.7%) | 34 (2.3%) | 1.156 | 0.693–1.928 | 0.579 |
| MACE | 105 (10.6%) | 126 (8.8%) | 1.231 | 0.937–1.617 | 0.134 |
| Major Bleeding | 323 (32.3%) | 588 (40.3%) | 0.704 | 0.595–0.833 | < 0.001 |

Values are n (%) or n, unless otherwise indicated.

AMT = aspirin monotherapy; DAPT = dual antiplatelet therapy; OR = odds ratio; CI = confidence interval

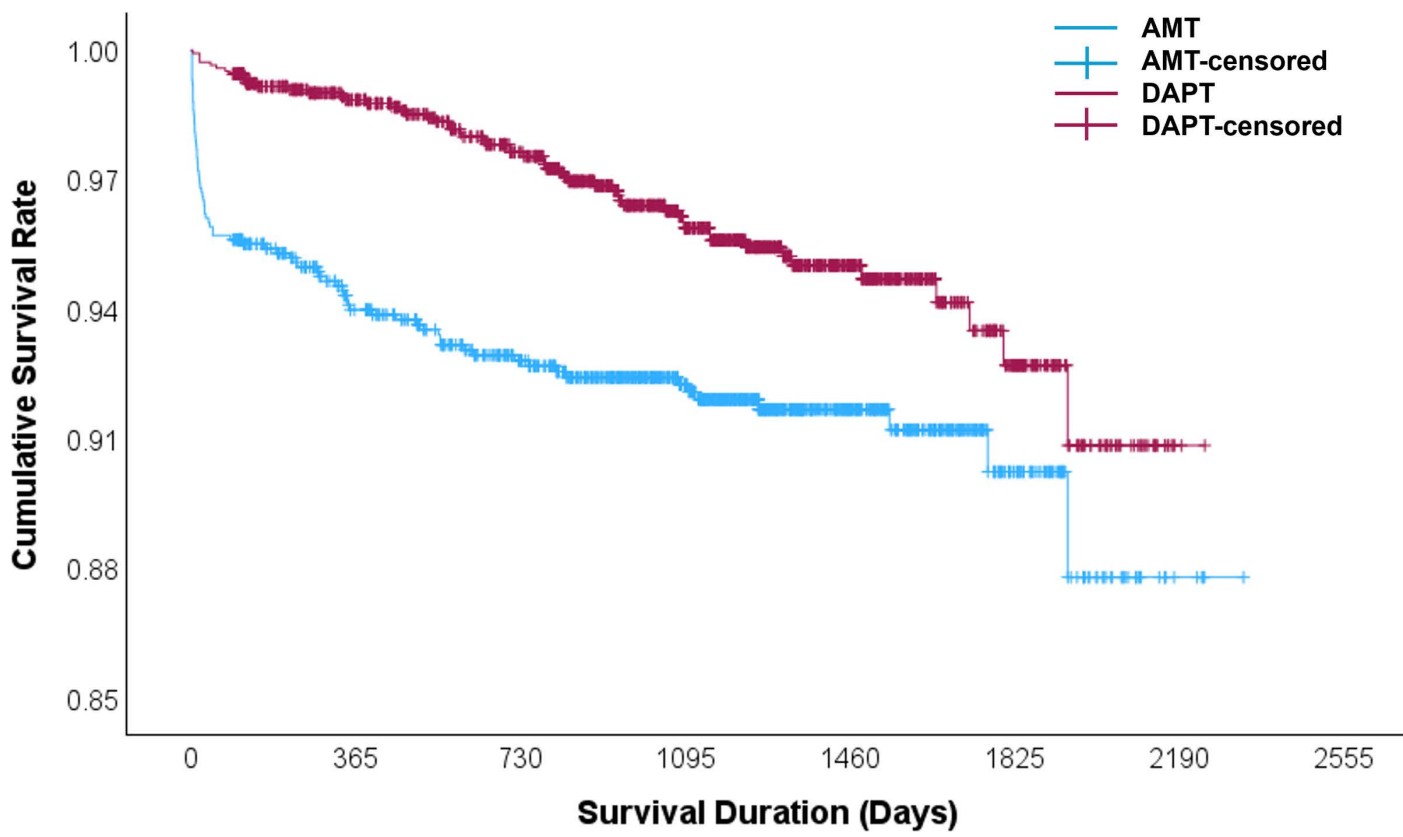

**Fig 1. Kaplan Meier curve.**

The model [0.062(BMI) - 1.255 (P2Y12 antagonist) - 1.160(GFR) - 0.039(EF) - 2.837] accounted for 14% of the variance in the SCD ($R^2 = 0.144$; $p < 0.001$).

Risk of SCD associated with each factor. The bars represent the hazard ratios (HR) for each factor, with error bars indicating the 95% confidence intervals. Red bars show an increase in risk, while green bars indicate a reduction in risk. The labels above each bar provide the percentage change in risk relative to an HR of 1, clarifying the degree of risk increase or reduction.

**Table 3. Predictors of SCD.**

| Covariates | Hazard Ratio | 95% CI | p-value |
|---|---|---|---|
| P2y12 Antagonist | 0.285 | 0.135–0.603 | 0.001 |
| BMI | 1.064 | 1.012–1.118 | 0.014 |
| GFR ≥ 60 (compared to GFR<60) | 0.314 | 0.158–0.624 | < 0.001 |
| EF | 0.962 | 0.939–0.986 | 0.002 |

The model [0.062*BMI - 1.255 (if DPAT) - 1.160*GFR - 0.039*EF - 2.837] accounted for 14% of the variance in the SCD ($R^2 = 0.144$; $p < 0.001$). A 1 unit increase in BMI is associated with a 6.4% increase in the risk of SCD, and a 1% increase in EF is associated with a 3.8% decrease in the risk of SCD.

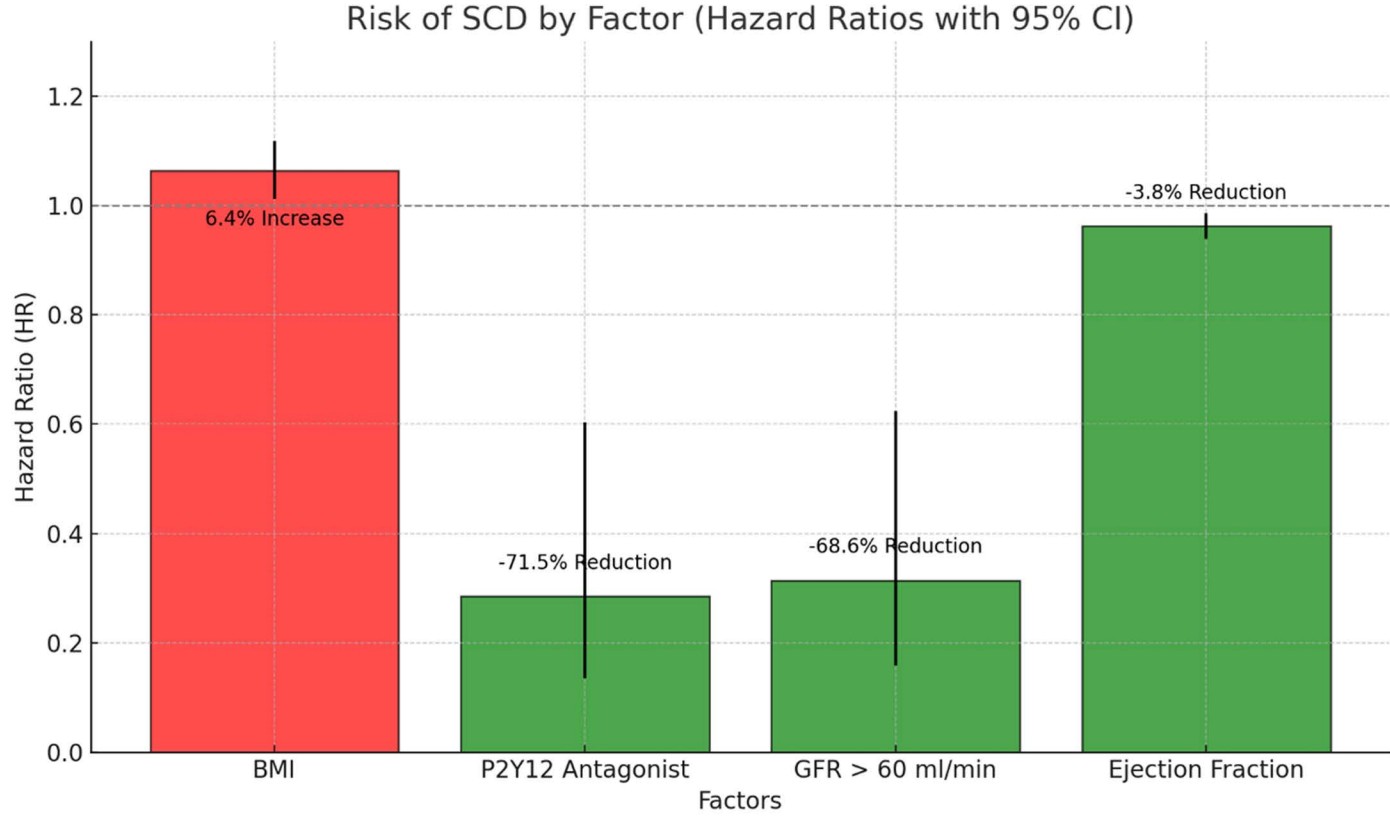

**Fig 2. Risk Illustration.**

**Table 4. Obesity and Depressed Ejection Fraction.**

| Covariates | Hazard Ratio | 95% CI | p-value |
|---|---|---|---|
| BMI > 30 (compared to BMI < 30) | N/S and thus not included in the model | | N/S |
| BMI > 35 (compared to BMI < 35) | 2.400 | 1.204–4.787 | 0.013 |
| EF > 35% (compared to EF < 35) | 0.354 | 0.139–0.904 | 0.030 |
| EF >40% (compared to EF < 40) | 0.347 | 0.158–0.763 | 0.008 |

Furthermore, compared to the patients with a BMI < 35, the patients with BMI ≥ 35 have a higher risk of SCD (HR = 2.40, 95% CI: 1.204–4.787; $p = 0.013$). Similarly, compared to patients with an EF < 35%, patients with an EF ≥ 35% have a lower risk of SCD (HR = 0.354, 95% CI0.139–0.904; $p = 0.03$)

## Discussions

### Patient characteristics and demographics

This study analyzed outcomes in 2,476 patients who underwent CABG and were subsequently treated with either AMT or DAPT (Fig 3). The findings provide significant insights into the role of DAPT in preventing SCD and other adverse events post-CABG. The data reveals distinct differences in patient characteristics between those on AMT and DAPT. Patients receiving AMT were healthier but had a higher proportion of patients reporting stable angina. Conversely, the DAPT group exhibited a higher prevalence of peripheral artery disease, reduced glomerular filtration rate (GFR < 60), prior PCI, pre-CABG use of DAPT, and a higher likelihood of undergoing on-pump surgery with a longer operative duration (see Table 1). These differences might indicate a more aggressive management approach in patients with more complex coronary disease or those at higher risk of thrombotic events.

### DAPT and sudden cardiac death after CABG

After adjusting for these baseline differences, the study found a significantly lower incidence of SCD in patients receiving DAPT compared to those on AMT (0.8% vs. 3.1%). This finding is critical, as it suggests that DAPT may offer substantial protective effects against SCD in the post-CABG setting. The adjusted odds ratio (OR = 3.831) indicates that patients on AMT were nearly four times more likely to experience SCD than those on DAPT, highlighting the potential benefit of DAPT beyond its established role (see Table 2). While several randomized trials and metanalyses have demonstrated significant survival benefits with the use of DAPT in patients undergoing CABG after ACS [10–14]. The benefit of DAPT has not been reported or highlighted as potential treatment to prevent risk of SCD after CABG. Currently most guidelines focus on conventional therapies and general guidelines for managing patients with CAD, such as lifestyle modification, medical management of cardiovascular risk factors. Post-hoc analysis reporting on SCD after CABG showed that the numerically greatest monthly rate of SCD was in the 31- to 90-day time period [2]. For those with depressed left ventricular ejection fraction, defibrillator is not recommended until 90 days after CABG. It is crucial to identify potential treatments that can lower the risk of SCD or any death in general in this time. Perhaps DAPT therapy is one such treatment strategy and merits further research to address this question more definitively. DAPT appears to reduce early mortality within the first 3 months post-CABG by preventing thrombotic graft occlusion, a key mechanism

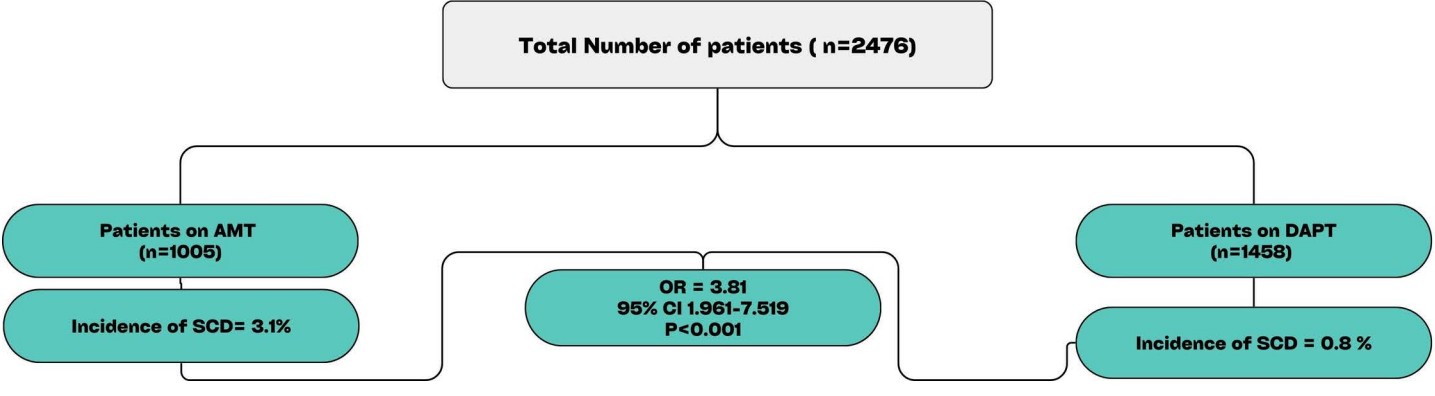

**Fig 3. Central illustration comparing sudden cardiac death incidence between AMT and DAPT groups.**

underlying the observed survival benefit and the primary objective of this treatment strategy. The AMT group demonstrated a lower incidence of major bleeding compared to the DAPT group (OR: 0.704, 95% CI: 0.595–0.833, p < 0.001), indicating a more favorable bleeding profile. However, there was no significant difference in transfusion requirements between the two groups, with rates of 9.9% in the AMT group and 15.6% in the DAPT group (OR: 1.11, 95% CI: 0.85–1.45, p = 0.48). Importantly, no cases of cardiac tamponade or CNS bleeding were reported in either group, though one patient in the DAPT group died due to bleeding complications. These findings highlight the generally acceptable safety profile of DAPT after CABG, despite a slightly higher bleeding risk.

## Binary regression model

The binary regression model further elucidates factors associated with a reduced risk of SCD, showing that the use of P2Y12 antagonists, better renal function (GFR > 60 mL/min), higher ejection fraction (EF), and lower body mass index (BMI) are all protective against SCD (Table 3). While obesity and depressed ejection fraction has been reported as predictors of SCD previously, this study presents some interesting insights relevant to this topic.

a) **Obesity:**

These findings of our study underscore the strong association between higher BMI and increased risk of SCD in post-CABG patients. Specifically, each $1 \, kg/m^2$ increase in BMI was linked to a 6.4% rise in SCD risk, highlighting the incremental effect of excess weight on adverse cardiovascular outcomes (Fig 2). Furthermore, patients categorized as morbidly obese (BMI > 35) were significantly more likely to experience SCD compared to those with a BMI below 35, with a hazard ratio of 2.4 (95% CI: 1.204–4.787; p = 0.013) (Table 4). Although there has been extensive research regarding the contribution of smoking to the pathogenesis of CAD and SCD the contribution of obesity is less understood. Blood volume and cardiac output increase with BMI, and obesity is known to affect diastolic function and to be associated with left ventricular hypertrophy (particularly when BMI is $> 30 \, kg/m^2$) and a prolonged QT-interval. This association suggests that extreme obesity may intensify underlying cardiovascular risks, likely through mechanisms such as increased inflammation, endothelial dysfunction, and heightened arrhythmic potential and place patients at much higher risk of SCD. These findings reinforce the need for targeted interventions to manage obesity in post-CABG patients and align with existing literature on obesity and its association with adverse cardiovascular outcomes [15–17]. Our study also contrasts with the obesity paradox theory [18], which suggests better survival in obese patients with acute cardiovascular decompensation.

b) **Ejection fraction:**

As previously reported, we observed a significant protective role of higher ejection fraction in reducing the risk of SCD following CABG. Specifically, each unit increase in EF was associated with a 3.8% reduction in SCD risk (HR = 0.962, 95% CI: 0.939–0.986, p = 0.002), suggesting that better cardiac function mitigates mortality risk in this patient population (Fig 2). Additionally, patients with an EF above 40% had significantly lower SCD incidence than those with an EF below 40% (HR = 0.347, 95% CI: 0.158–0.763; p = 0.008) (Table 4). This marked reduction underscores that preserved left ventricular function may act as a critical determinant of post-CABG survival. The findings align with existing evidence indicating that reduced EF heightens vulnerability to arrhythmias and other adverse events. Emphasizing ejection fraction as a risk stratifier could enhance clinical decisions regarding post-CABG management, guiding interventions such as medical optimization or monitoring to support improved

long-term outcomes. While patients with persistent low ejection fraction despite optimal GDMT are recommended to have ICD implant [9]; it is not recommended for patients undergoing CABG as a prophylaxis. Perhaps more research and evidence are needed to address these important questions.

### c) Chronic kidney disease:

Higher renal function, as indicated by a glomerular filtration rate (GFR) above 60 ml/min, is associated with a significantly lower risk of sudden cardiac death (SCD) following CABG (Fig 2). The binary regression model revealed that patients with GFR > 60 ml/min were approximately 69% less likely to experience SCD compared to those with lower GFR values (HR = 0.314, 95% CI: 0.158–0.624; p < 0.001) (Table 4). This association highlights the importance of preserved kidney function in improving survival outcomes post-CABG. Impaired renal function is known to exacerbate cardiovascular risk by contributing to chronic inflammation, oxidative stress, and dysregulation of electrolytes, which can increase susceptibility to arrhythmias and SCD [19].

### d) P2Y12 antagonist:

An HR of 0.285 indicates a 71.5% reduction in the risk of SCD for patients prescribed P2Y12 antagonists (1 − 0.285 = 0.715). This strong reduction effect is supported by a highly significant p-value (p < 0.001) and a narrow CI (0.135–0.603), which suggests robust evidence that P2Y12 antagonist use is protective against SCD (Fig 2). Kulik et al. demonstrated a decreased incidence of SCD with ticagrelor compared to clopidogrel, suggesting that the prescription of P2Y12 antagonist after CABG can impact mortality outcomes [11]. As novel P2Y12 inhibitors, such as ticagrelor, continue to yield superior clinical outcomes in managing chronic coronary artery disease, there is a growing expectation of similar benefits in CABG patients [12,13]. Our study's findings underscore the need for further research to confirm the potential advantages of P2Y12 antagonists specifically within the post-CABG population.

Among these factors, prescription of P2Y12 antagonists and GFR > 60 ml/min demonstrate the greatest reduction in SCD risk. Higher ejection fraction also contributes to risk reduction, though to a lesser degree, while higher BMI is associated with increased risk. Each factor shows a statistically significant relationship with SCD risk, highlighting their importance in patient management following CABG.

This study reinforces previous findings that AMT carries a higher risk of all-cause and cardiovascular mortality than DAPT, with ORs of 2.136 and 3.731, respectively (Fig 1). It highlights the need to explore novel pharmacotherapies and understand their impact on survival, underlying mechanisms, and clinical implications. Current literature and guidelines primarily focus on established medical therapies, with limited attention to newer antiplatelet agents and their role in preventing SCD. While guidelines recommend defibrillator placement in chronic heart disease patients with reduced ejection fraction to prevent SCD, patients who undergo CABG with low ejection fraction must wait 90 days post-surgery to qualify for a defibrillator [20–22], during which time guideline-directed medical therapy is emphasized to improve cardiac function [23–27]. These patients face risks of fatal arrhythmic events within the first 90 days and SCD from non-arrhythmic mechanisms, thereafter, posing a significant challenge. With novel antiplatelet therapies emerging in post-CABG care [20–22], pharmacotherapies such as DAPT hold promise as strategies to mitigate SCD risk in CABG patients. Our findings support DAPT's potential role in improving graft patency and preventing coronary ischemia that can lead to fatal arrhythmias, particularly in high-risk patients. Given DAPT's substantial impact on mortality outcomes, further studies are warranted to refine patient selection and optimize DAPT duration in the post-CABG setting.

Table 5. Causes of sudden cardiac death in AMT vs DAPT group.

| Causes | DAPT | AMT |
|---|---|---|
| Ventricular tachycardia | 1 | 7 |
| Ventricular fibrillation | 4 | 5 |
| Pulseless electrical activity/ Asystole | 2 | 4 |
| Unwitnessed sudden death | 4 | 24 |
| Other | 1 | 0 |
| Total | 12 | 40 |

## Limitations and future research

Some of the unwitnessed deaths could have been due to non-cardiac causes and we cannot for certain say that these deaths are cardiac. Judgement was made by chart review, death certificate and in some case by phone interview with family member. The different causes of sudden cardiac death are outlined in Table 5. This study's retrospective design and single-center setting may limit the generalizability of the findings. A notable limitation of the study is the unavailability of lipid profile data, which prevented the analysis of its potential impact on outcomes. Another limitation of this study is that, apart from antiplatelet therapy, which was carefully verified for both type and duration of treatment, most other medications, such as statins and beta blockers, were recorded based on the discharge medication list at the time of surgery. Consequently, there may be cases where patients initially prescribed lower-intensity statins were later switched to high-intensity therapy, or where patients were not prescribed statins or had their statin therapy discontinued due to intolerance. Additionally, observational design, the lack of matching between groups precludes establishing a direct cause-and-effect relationship between DAPT and clinical outcomes such as survival and complete revascularization.

Future studies should focus on evaluating the specific role of DAPT in SCD prevention, with an emphasis on tailoring therapy to individual patient risk profiles and optimizing treatment duration.

## Conclusion

In conclusion, DAPT presents potential benefits in reducing SCD and other adverse outcomes post-CABG, individual patient risk factors such as bleeding risk, renal function, BMI, and left ventricular function must be carefully considered when devising a comprehensive, personalized post-operative care plan.

**Tweet:** In a retrospective observational study evaluating antiplatelet therapy after CABG, the use of DAPT decreased this risk of SCD compared to AMT.

## Author contributions

**Conceptualization:** Iftikhar Ali Ch, Naeem Tahirkheli.

**Data curation:** Iftikhar Ali Ch, Pei-Tzu Wu, Mashal Tahirkheli, Rahat Jamal, Sabrina Chaudry, Ananya Bhaktaram, Faris Amil.

**Formal analysis:** Iftikhar Ali Ch, Azhar Chaudhry, Mishal Zehra, Pei-Tzu Wu.

**Investigation:** Iftikhar Ali Ch, Azhar Chaudhry, Mishal Zehra, Mashal Tahirkheli, Poonam Bai, Steven Miller, Jeffrey Garrett, Naeem Tahirkheli.

**Methodology:** Iftikhar Ali Ch, Pei-Tzu Wu, Rahat Jamal.

**Project administration:** Iftikhar Ali Ch, John Randolph.

**Resources:** Rahat Jamal, Sabrina Chaudry.

**Supervision:** Jeffrey Garrett, Naeem Tahirkheli.

**Validation:** Iftikhar Ali Ch, John Randolph, Steven Miller, Jeffrey Garrett, Naeem Tahirkheli.

**Writing – original draft:** Iftikhar Ali Ch, Poonam Bai.

**Writing – review & editing:** Iftikhar Ali Ch, Azhar Chaudhry, Mishal Zehra, Poonam Bai, John Randolph.

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
