## [Decision Letter · Decision Letter 0]

5 Jan 2025

PONE-D-24-50328Sudden Cardiac Death After Coronary Artery Bypass Graft Surgery and Role of Antiplatelet TherapyPLOS ONE

Dear Dr. Ch,

Thank you for submitting your manuscript to PLOS ONE. After careful consideration, we feel that it has merit but does not fully meet PLOS ONE’s publication criteria as it currently stands. Therefore, we invite you to submit a revised version of the manuscript that addresses the points raised during the review process.

We look forward to receiving your revised manuscript.

Kind regards,

Chiara Lazzeri

Academic Editor

PLOS ONE

Journal Requirements:

2. In the online submission form, you indicated that “Data can be made available upon request.”

5. Please ensure that you refer to Figure 1-3 in your text as, if accepted, production will need this reference to link the reader to the figure.

6. We note you have included a table to which you do not refer in the text of your manuscript. Please ensure that you refer to Table 1, 3, 4 and 5 in your text; if accepted, production will need this reference to link the reader to the Table.

Reviewers' comments:

Reviewer's Responses to Questions

**Comments to the Author**

1. Is the manuscript technically sound, and do the data support the conclusions?

Reviewer #1: Yes

Reviewer #2: Yes

2. Has the statistical analysis been performed appropriately and rigorously? 

Reviewer #1: Yes

Reviewer #2: Yes

3. Have the authors made all data underlying the findings in their manuscript fully available?

Reviewer #1: Yes

Reviewer #2: Yes

4. Is the manuscript presented in an intelligible fashion and written in standard English?

Reviewer #1: Yes

Reviewer #2: Yes

5. Review Comments to the Author

Reviewer #1: I thank the authors for the quality of the manuscript providing an interesting single-center analysis upon a well-known topic, which’s characteristics, physiopathology and pharmacology prevention are far from being completely understood.

The manuscript is written in good English without major grammatical errors.

No problems emerged in the structure of the manuscript.

The similarity index is close to 3% with no major limitations for publishing.

Limitations could be represented by the retrospective form of the study, the single center experience and some possible bias about the correct identification of SCD in AMT group correctly defined and described in the manuscript.

Other possible limitations can be represented by the absence of groups matching between the AMT and DAPT group.

The statistical analysis is correctly mad. No specific suggestions to report.

The results found with statistical analysis support the conclusion made by the colleagues with HR, p values and confidence interval correctly reported and enlightened in the manuscript.

I’d suggest magnifying the tables and graphics references in the results part if the manuscript, permitting the reader to rapidly identify the references.

For these reasons I’d suggest a minor review before publication.

Reviewer #2: The authors performed a retrospective observational study as “Sudden Cardiac Death After Coronary Artery Bypass Graft Surgery and Role of Antiplatelet Therapy” including total of 2,476 patients followed in this post-CABG study, the analysis included 1,005 patients who received

aspirin monotherapy (AMT) and 1,458 patients who received dual antiplatelet therapy (DAPT). They concluded that DAPT prescription after CABG improves survival and lowers risk of sudden cardiac death.

The topic is currently interesting and I think this study and article had good clinical value.

Comments:

1. This study concluded DAPT presents potential benefits in reducing SCD post-CABG. My main questions are what kinds of SCD reduced by DAPT and the mechanism? In comparison of AMT, the main benefit of DAPT can possibly reduce acute major adverse cardiac and cerebrovascular events (MACCE), defined as a composite of death, myocardial infarction, stroke, or repeat revascularization but not death caused by heart failure and arrhythmia. In a previous propensity-matched analysis, DAPT did not confer any advantage in terms of improved survival or freedom from MACCE compared to aspirin monotherapy following isolated CABG(1). In this study, the authors didn’t demonstrate the difference of Post-CABG acute coronary syndrome (ACS) and myocardial infarction(MI) between AMT and DAPT groups, which are possible main reasons of reducing sudden cardiac death by DAPT and P2Y12 antagonist.

2. In addition, the level of low density lipoprotein (LDL) and the significant different proportion of statin and AAD administration are also important confounding factors for sudden cardiac death, I suggest listed it and do univariate and multivariate cox regression analysis.

(1)Reference: Hess NR, Sultan I, Wang Y, Thoma F, Kilic A. Comparison of Aspirin Monotherapy versus Dual Antiplatelet Therapy Following Coronary Artery Bypass Grafting. Am J Cardiol. 2021 Jun 1;148:44-52. doi: 10.1016/j.amjcard.2021.02.026. Epub 2021 Mar 3. PMID: 33667447.

6. PLOS authors have the option to publish the peer review history of their article (what does this mean?). If published, this will include your full peer review and any attached files.

Reviewer #1: **Yes: **prof. Pierluigi Stefano Director Cardiac Surgery Department Careggi University Hospital

Reviewer #2: No

---

## [Author Response · Author response to Decision Letter 0]

27 Jan 2025

Dear Editor and Reviewers,

We sincerely thank you for your detailed and constructive feedback, which has been invaluable in improving the clarity and rigor of our manuscript. Below, we address each reviewer’s comments and outline the revisions made. We did not identify any retracted studies among the cited references at this time. However, please let us know if we have inadvertently overlooked any retracted references.

Reviewer 1

Q1:

I’d suggest magnifying the tables and graphics references in the results part of the manuscript, permitting the reader to rapidly identify the references.

Response:

Thank you for this valuable suggestion. We have revised the manuscript to ensure that references to tables and graphics in the results section are now more prominent and easily identifiable.

Reviewer 2

Q2:

What kinds of sudden cardiac death (SCD) are reduced by DAPT, and what is the mechanism? In comparison to aspirin monotherapy (AMT), the main benefit of DAPT can possibly reduce acute major adverse cardiac and cerebrovascular events (MACCE), defined as a composite of death, myocardial infarction, stroke, or repeat revascularization, but not death caused by heart failure and arrhythmia. In a previous propensity-matched analysis, DAPT did not confer any advantage in terms of improved survival or freedom from MACCE compared to aspirin monotherapy following isolated CABG.

Response:

We appreciate the reviewer’s insightful comments. DAPT appears to reduce early mortality within the first 3 months post-CABG by mitigating thrombotic graft occlusion, a critical mechanism underlying the observed survival benefit. Unlike the referenced study, which relied solely on discharge records and didn’t ensure compliance, we manually reviewed patient charts and confirmed antiplatelet prescription compliance during follow-ups. Additionally, we employed stricter and clinically relevant exclusion criteria, excluding only deaths within 48 hours post-surgery, whereas the referenced study excluded all in-hospital deaths, potentially underestimating DAPT's benefits. While Hess et al. did not demonstrate significant MACCE benefits, their data showed a trend toward better survival with DAPT. Current ACC/AHA guidelines support DAPT post-CABG in ACS patients, aligning with our findings of improved early postoperative survival. We have clarified this in the revised manuscript. Additionally in our discussion we have provided citations to several studies proving survival benefit of DAPT after CABG including study by Sorenson et al and various other studies (refer to citations 10-14) have shown promising results with DAPT treatments.

Reference: Sørensen R, Abildstrøm SZ, Hansen PR, Hvelplund A, Andersson C, Charlot M, Fosbøl EL, Køber L, Madsen JK, Gislason GH, Torp-Pedersen C. Efficacy of post-operative clopidogrel treatment in patients revascularized with coronary artery bypass grafting after myocardial infarction. Journal of the American College of Cardiology. 2011 Mar 8;57(10):1202-9.

Q3:

In this study, the authors didn’t demonstrate the difference in post-CABG acute coronary syndrome (ACS) and myocardial infarction (MI) between the AMT and DAPT groups, which are possible main reasons for reducing SCD by DAPT and P2Y12 antagonists.

Response:

We agree with the reviewer that sudden cardiac deaths due to VT/VF or unwitnessed SCD are likely attributable to fatal MI and/or ischemic events. However, in our study, no significant differences were observed between the AMT and DAPT groups in the incidence of non-fatal MI, CVA, or the composite of these events as MACE.

Q4:

The level of low-density lipoprotein (LDL) and the significant proportion of statin and anti-arrhythmic drug (AAD) administration are also important confounding factors for SCD. I suggest listing these and performing univariate and multivariate Cox regression analysis.

Response:

We conducted multivariate analysis and found that neither statins nor AADs showed significant protective effects warranting inclusion in the final hazard ratios. We focused on clinically meaningful covariates in the analysis. Unfortunately, DAPT appears to reduce early mortality within the first 3 months post-CABG by preventing thrombotic graft occlusion, a key mechanism underlying the observed survival benefit and the primary objective of this treatment strategy

We have added the following statement to the manuscript:

“Common post-CABG medications, including ACE inhibitors/ARBs, beta-blockers, antiarrhythmic drugs (AAD), calcium channel blockers (CCBs), and statins, were not significant predictors of SCD (p=0.380, 0.450, 0.428, 0.337, and 0.185, respectively).”

The role of statins in improving overall survival (but not SCD) post-CABG is addressed in a separate article under review.

Also we have added a statement under limitations, “Another limitation of this study is that, apart from antiplatelet therapy, which was carefully verified for both type and duration of treatment, most other medications, such as statins and beta blockers, were recorded based on the discharge medication list at the time of surgery. Consequently, there may be cases where patients initially prescribed lower-intensity statins were later switched to high-intensity therapy, or where patients were not prescribed statins or had their statin therapy discontinued due to intolerance.”

Limitations

Reviewer 1:

Suggested limitations have been added.

Response:

We appreciate the reviewer’s comments and have incorporated the suggested limitations into the revised manuscript. These include the retrospective nature of the study, single-center experience, and potential bias in correctly identifying SCD in the AMT group. Additionally, the absence of propensity matching between the AMT and DAPT groups has been noted as a limitation.

Thank you for your constructive feedback, which has strengthened our manuscript.

Sincerely,

Iftikhar Ali Ch, MD

South Oklahoma Research

5200 E Interstate 240 Service Road, Oklahoma City, OK 73135

Tel: 405-628-6000 | Fax Number: 405-628-0000 | E-mail: babar_175@hotmail.com

---

## [Editor Report · Decision Letter 1]

30 Jan 2025

Sudden Cardiac Death After Coronary Artery Bypass Graft Surgery and Role of Antiplatelet Therapy

PONE-D-24-50328R1

Dear Dr. Ch,

We’re pleased to inform you that your manuscript has been judged scientifically suitable for publication and will be formally accepted for publication once it meets all outstanding technical requirements.

Kind regards,

Chiara Lazzeri

Academic Editor

PLOS ONE
---

## [Editor Report · Acceptance letter]

PONE-D-24-50328R1

PLOS ONE

Dear Dr. Ch,

I'm pleased to inform you that your manuscript has been deemed suitable for publication in PLOS ONE. Congratulations! Your manuscript is now being handed over to our production team.

Kind regards,

on behalf of

Dr. Chiara Lazzeri

Academic Editor

PLOS ONE